

# Evaluation of putative reference genes for quantitative real-time PCR normalization in *Lilium regale* during development and under stress

Qiang Liu*, Chi Wei*, Ming-Fang Zhang and Gui-Xia Jia

Beijing Key Laboratory of Ornamental Plants Germplasm Innovation & Molecular Breeding, National Engineering Research Center for Floriculture, Beijing Laboratory of Urban and Rural Ecological Environment and College of Landscape Architecture, Beijing Forestry University, Beijing, China
* These authors contributed equally to this work.

## ABSTRACT

Normalization to reference genes is the most common method to avoid bias in real-time quantitative PCR (qPCR), which has been widely used for quantification of gene expression. Despite several studies on gene expression, *Lilium*, and particularly *L. regale*, has not been fully investigated regarding the evaluation of reference genes suitable for normalization. In this study, nine putative reference genes, namely *18S rRNA*, *ACT*, *BHLH*, *CLA*, *CYP*, *EF1*, *GAPDH*, *SAND* and *TIP41*, were analyzed for accurate quantitative PCR normalization at different developmental stages and under different stress conditions, including biotic (*Botrytis elliptica*), drought, salinity, cold and heat stress. All these genes showed a wide variation in their Cq (quantification Cycle) values, and their stabilities were calculated by geNorm, NormFinder and BestKeeper. In a combination of the results from the three algorithms, *BHLH* was superior to the other candidates when all the experimental treatments were analyzed together; *CLA* and *EF1* were also recommended by two of the three algorithms. As for specific conditions, *EF1* under various developmental stages, *SAND* under biotic stress, *CYP/GAPDH* under drought stress, and *TIP41* under salinity stress were generally considered suitable. All the algorithms agreed on the stability of *SAND* and *GAPDH* under cold stress, while only *CYP* was selected under heat stress by all of them. Additionally, the selection of optimal reference genes under biotic stress was further verified by analyzing the expression level of *LrLOX* in leaves inoculated with *B. elliptica*. Our study would be beneficial for future studies on gene expression and molecular breeding of *Lilium*.

# INTRODUCTION

As one of the standard methods for gene expression profiling, real-time quantitative PCR (qPCR) is featured with a high sensitivity, specificity and throughput in comparison with other previous molecular techniques (*Gachon, Mingam & Charrier, 2004*). During

Corresponding author
Gui-Xia Jia, gxjia@bjfu.edu.cn

the qPCR assay, data normalization is a prerequisite for accurate and reproducible measurement of quantitative expression. A common strategy for normalization is to normalize the mRNA expression of the gene of interest to a reference gene. Accordingly, the accuracy of the measurement is directly relied on the expression stability of the reference gene in different samples, which may vary in tissues or organs, developmental stages or the experimental conditions (*Radonić et al., 2004*). A number of housekeeping genes are commonly used as reference genes, such as *Glyceraldehyde-3-Phosphate Dehydrogenase* (*GAPDH*), *18S ribosomal RNA* (*18S rRNA*), *Elongation Factor 1-α* (*EF1*) and *Actin* (*ACT*) (*Czechowski et al., 2005*; *Gutierrez et al., 2008*). These genes are involved in basic and ubiquitous cellular processes including glycolytic metabolism, ribosome biosynthesis, cytoskeleton constitution and protein folding. It is assumed that such genes are expressed at a constant level in all tissues independently of growing environment (*Dheda et al., 2004*). However, it is evidenced that the transcription levels of these housekeeping genes can be unstable in response to different experimental conditions or in different tissue types, leading to errors in quantification of target genes (*Nicot, 2005*; *Schönbach et al., 2012*). Therefore, it is necessary to verify the expression stability of candidate reference genes across various experimental treatments or tissue types (*Bin et al., 2012*; *Nicot, 2005*). Several statistical algorithms, including geNorm (*Vandesompele et al., 2002*), NormFinder (*Andersen, Jensen & Orntoft, 2004*) and BestKeeper (*Pfaffl et al., 2004*), were introduced to identify the most suitable gene among a set of candidates for qPCR normalization. Based on these algorithms, evaluation of putative reference genes has been performed on a number of plant species (*Bin et al., 2012*; *Fu et al., 2012*; *Nicot, 2005*; *Schönbach et al., 2012*).

As a bulbous flowering plant with great ornamental value, *Lilium* is one of the most important floricultural crops in the world. Thousands of cultivars in this genus were produced under inter-specific hybridization and artificial selection in the last twenty years. However, only a limited number of studies have been carried out on the gene expression of *Lilium*, and the reference genes used in these studies, namely *ACT*, *18S rRNA* or *GAPDH* (*He et al., 2014*; *Li et al., 2014*; *Tong et al., 2013*; *Wang et al., 2009*; *Xin et al., 2010*), was only based on the results from other species without an experimental comparison among the candidate reference genes of *Lilium*. Only until very recently the evaluation of reference genes in *L. davidii* var. *unicolor* and *L. brownii* has been reported (*Li et al., 2015*; *Luo et al., 2014*), but specific studies on these genes under certain conditions, such as different developmental stages or stresses, are still needed for more accurate normalization in qPCR. As a promising material for breeding and gene study in section Leucolirion of *Lilium*, *L. regale* is a wild lily species distributing across Min River basin in Sichuan Province of China. It was found to possess extremely high resistance to biotic and abiotic stresses (*Zhao et al., 1994*). Studies on its reference genes under environmental stresses would be essential for analyzing its expression pattern and understanding the mechanism of stress resistance, and providing guidance to the molecular breeding of lilies.

Because the knowledge on gene expression profiling in *Lilium* is still limited, transcriptome and Digital Gene Expression (DGE) technology would play a pivotal

role in the reference gene selection. Therefore, in this study, the expression stability of nine putative reference genes were evaluated under different developmental stages and various stress treatments. Three statistical algorithms, geNorm, NormFinder and BestKeeper, were combined to analyze the expression stabilities of these candidate reference genes.

## MATERIALS AND METHODS

### Plant materials and sampling

Bulbs of *L. regale* with a circumference of 120 mm were stored at 4 °C for 3 months, and then planted in pots (diameter 100 mm) filled with substrate (sterile turf: vermiculite: perlite = 1:1:1, v/v/v). They were grown in a growth chamber with 12 h light (200 $\mu$mol $\cdot$ m$^{-2}$ $\cdot$ s$^{-1}$)/12 h dark photoperiod and 25 °C day/22 °C night thermoperiod, at a relative humidity of 80%. The plants were daily watered to maintain the field capacity to the maximum. For quantification at different developmental stages, one leaf of each plant was sampled every month after germination. The sampling was conducted for six months from the fifth to tenth leaf below the apex, throughout the vegetative and reproductive stages of the plants before they entered senescence.

The plants generated from the bulbs were used for other treatments, which were conducted during the flower bud stage. For drought stress treatment, the water supply was withheld and leaf samples were collected at 2–4, 6 and 8 d during the treatment. For salinity stress treatment, the pots with plants were completely saturated with water containing 100 mM NaCl, followed by leaf sampling at 2, 6, 12, 24, 36 and 48 h during the treatment. For heat or cold stress treatment, the pots with plants were transferred to another growth chamber with a same photoperiod at 37 °C (heat stress) or 4 °C (cold stress). Leaves from the treated plants were collected at 2, 6, 12, 24, 36 and 48 h during the treatment. One leaf of each plant was sampled at every time point from the fifth to tenth leaf below the apex.

For biotic stress treatment, the fifth to tenth leaf below the apex of each plant were detached and surface-sterilized by 0.1% sodium hypochlorite for inoculation. *Botrytis elliptica* was cultured on Potato Dextrose Agar medium (PDA) under near-UV light for 5 days. The conidia were collected by successive gentle vortexing in Tween 20 solution (0.05% Tween 20 in sterile deionized water). After adjusted to $5 \times 10^4$ conidia $\cdot$ mL$^{-1}$, the suspension was sprayed on the abaxial surface of the leaves. Mock inoculations with water were also performed as a control group. The petioles of inoculated leaves were covered with cotton infiltrated by fertilized water, and the leaves were incubated on wet filter papers in petri dishes, which were placed in a growth chamber with 100% relative humidity, 12 h light (200 $\mu$mol $\cdot$ m$^{-2}$ $\cdot$ s$^{-1}$)/12 h dark photoperiod and 25 °C day/22 °C night thermoperiod. Samples were collected at 2, 6, 12, 24, 36 and 48 hpi (hours post inoculation).

All the samples collected were immediately frozen in liquid nitrogen and stored at −80 °C. Two biological replications were designed for every developmental stage or every time point during the stress treatments, each replication consisting of three leaves from three individual plants as a sample pool. Twelve sample pools from each treatment

(except 10 from drought treatment), namely 70 sample pools from 210 plants in total were used for study.

## Total RNA extraction and cDNA synthesis

Total RNA from each sample was extracted using the EASYspin Plus Plant RNA kit (RN38, Aidlab Biotech, Beijing, China) according to the manufacturer's protocol. Each RNA sample was treated with an RNase-free DNase I (Promega, Madison, WI, USA) at 37 °C for 30 min to eliminate the trace contaminants of genomic DNA. The concentration and integrity of the samples were examined by NanoDrop 2000 spectrophotometer (Thermo Fisher Scientific, Waltham, MA, USA) and 1.4% agarose gel electrophoresis. The first strand cDNA was synthesized based on 1 μg total RNA of each sample using the M-MLV reverse transcription system (Promega) according to the manufacturer's protocol.

## Primer design and PCR confirmation

Primers were designed using Beacon designer 7 (Premier Biosoft, Palo Alto, CA, USA) following the stringent criteria: PCR product size between 75–200 bp, primer length between 18–25 bp, optimal $T_m$ between 55–65 °C, GC content between 40–60%. The theoretical $T_m$ of the amplicon for each gene were predicted by Primer Premier 5 (Premier Biosoft), in which the salt concentration was set as 50 mM. In order to check the specificity and suitable reaction condition of the primers, they were initially tested in a general PCR reaction with gradient temperature (52–60 °C) using cDNA as a template. For further confirmation, PCR products were cloned into pGEM-T Easy vectors (Promega) respectively and then sequenced by Sangon Biotech (Shanghai, China).

## qPCR analysis and efficiency

Real-time quantitative PCR was conducted with MiniOpticon Real-time PCR System (Bio-Rad, Hercules, CA, USA), using SYBR Premix Ex Taq (Takara Bio, Kusatsu, Shiga, Japan) to monitor dsDNA synthesis. It was carried out in a 20 μl mixture containing 2 μl cDNA template, 10 μl 2× SYBR Premix Ex Taq, and each primer in the optimized concentration shown in Table 1, by the following thermal cycles: 95 °C for 2 min, followed by 40 cycles of 95 °C for 5 s, the optimized annealing temperature shown in Table 1 for 10 s and 72 °C for 15 s. In order to ensure the specificity of each primer pair, melting curves were recorded after Cycle 40 by heating from 65–95 °C stepwise by 0.5 °C every 5 s. Each qPCR reaction was performed in technical triplicate.

   The reaction condition including the best primer concentration and annealing temperature was optimized using the cDNA sample pool of the total samples used in this study as templates, which were analyzed as five serial ten-fold dilutions. The amplification efficiency of each primer pair was derived from a standard curve generated after the optimization. Mean quantification Cycle (Cq) values of each ten-fold dilution were plotted against the logarithm of the cDNA dilution factor (*Ginzinger, 2002*).

## Data analysis

The Cq values for each tested reference gene were analyzed by a specific threshold. The raw Cq values were converted into relative quantities using the formula $Q = E^{-\Delta Cq}$, where $E$

**Table 1 Characteristics of the nine candidate reference genes.**

| Gene | Full name | Accession number | Cellular function |
|------|-----------|------------------|-------------------|
| 18S rRNA | 18S ribosomal RNA | HQ686070 | Ribosome subunit |
| ACT | Beta Actin | KJ543460 | Cytoskeletal structural protein |
| BHLH | Basic helix-loop-helix | KJ543467 | Transcription factor |
| CLA | Clathrin | KJ543465 | Clathrin adaptor complex |
| CYP | Cycolphilin A | KJ543464 | Serine-threonine phosphatase inhibitor |
| EF1 | Elongation factor 1-a | KJ543461 | Eukaryotic elongation factor 1 |
| GAPDH | Glyceraldehyde-3-phosphate dehydrogenase | KJ543468 | Oxidoreductase in glycolysis and gluconeogenesis |
| SAND | SAND family protein | KJ543463 | Hypothetical proteins |
| TIP41 | TIP41 family protein | KJ543466 | Tonoplast intrinsic proteins |

is the efficiency of the gene amplification for each primer pair and $\Delta Cq$ is the lowest Cq value as calibrator (which corresponds to the sample with the highest expression) minus the Cq value of the sample tested. The data obtained for each primer were analyzed using geNorm V3.5 (*Vandesompele et al., 2002*), NormFinder (*Andersen, Jensen & Orntoft, 2004*) and BestKeeper (*Pfaffl et al., 2004*) for reference gene selection.

## Normalization of *LrLOX*

The gene *LrLOX* (GenBank accession number KM051414), a putative homolog of *Lipoxygenases* (*LOXs*) in *L. regale*, was used to validate the selected reference genes. The primers were designed using Beacon designer 7 (Forward: 5′-TTCCAGCGACAACAGGAGCAC-3′; Reverse: 5′-CGTCGTCCACCAAATCCACTT-3′). The leaves of *L. regale* were inoculated with *B. elliptica* using the method mentioned above. The expression profiling of *LrLOX* in the samples collected at 2, 6, 12, 24, 36 and 48 hpi was analyzed under the same experimental procedure as above. The samples at 0 hpi were taken as control group. For comparative purposes, the relative expression of the target gene was calculated with different normalization factors based on the geometric mean of the three most stable genes and the most unstable genes under biotic stress, as well as the most stable genes in all conditions concluded from the analysis. The relative expression was calculated by standard $E^{-\Delta\Delta Cq}$ method. The data collected were subjected to one-way ANOVA and the means were analyzed by Tukey's range test. Data were represented as mean ± SD.

## RESULTS

### Selection of candidate reference genes and amplification specificity

In our previous study, a cDNA library of *L. regale* was constructed to analyze the gene expression pattern after inoculation with *B. elliptica* (Q Cui et al., 2015, unpublished data). Based on the RPKM (Reads per Kb per Million reads) values of the unigenes in the library (Table S1), as well as the previous studies on reference gene selection in other plants (*Bin et al., 2012*; *Fu et al., 2012*), a total of nine genes associated with a wide variety of biological functions were selected as candidate reference genes during development or

**Table 2 Primer sequences and optimized reaction conditions of the nine candidate reference genes.**

| Gene | Prime sequence (5′-3′) | Amplicon size (bp) | Primer concentration (μM) | $T_a$ (°C) | Amplicon $T_m$ (°C) | PCR efficiency (%) | Regression coefficient ($R^2$) |
|---|---|---|---|---|---|---|---|
| 18S rRNA | F: GCAGAATCCCGTGAACCAT | 143 | 0.3 | 54 | 87.8 | 93.7 | 0.998 |
| | R: GCCAATCTCCGCATCCAT | | | | | | |
| ACT | F: CCCATTGAGCACGGCATTGTC | 128 | 0.2 | 56 | 85.7 | 102.8 | 0.999 |
| | R: GGATTGAGAGGAGCTTCGGTGAGA | | | | | | |
| BHLH | F: CCAGCAGGTTGTCCTTGTG | 142 | 0.3 | 56 | 85.1 | 97.5 | 0.998 |
| | R: TCCGTGATGAGAAGCAGAGG | | | | | | |
| CLA | F: GATGAGATTCTGATTGCTGGTGAG | 103 | 0.3 | 55 | 84.2 | 93 | 0.999 |
| | R: CCTGCTCTTTGGCTGTTTCC | | | | | | |
| CYP | F: ACCCTTGGGCAAGAACAACAGAA | 127 | 0.3 | 56 | 84.3 | 99.3 | 0.997 |
| | R: GCAAAGGAGGTTGAGTTGGAGGAT | | | | | | |
| EF1 | F: GGCACTAACTCGCTCCTTCTG | 173 | 0.2 | 55 | 84.6 | 101.6 | 0.998 |
| | R: TTGGTAAGATGCTGGTGATTGGAT | | | | | | |
| GAPDH | F: CACGGTCAGTGGAAGCACCATGAGAT | 180 | 0.3 | 60 | 86.3 | 97.6 | 0.995 |
| | R: AGCAGCAGCCTTATCCTTGTCAGTGA | | | | | | |
| SAND | F: CCAATACCCAGATGAGGAGACAAA | 178 | 0.1 | 54 | 83.8 | 105.4 | 0.992 |
| | R: GGATTCGCATTGAGGCTGTTC | | | | | | |
| TIP41 | F: CGAAGCCAGAAACGGAGAAGAAT | 192 | 0.3 | 55 | 82.9 | 93.6 | 0.998 |
| | R: GGGTAGGGTGGATTGGGAAGA | | | | | | |

**Note:**
F, Forward; R, Reverse; $T_a$, Annealing temperature; Amplicon $T_m$, Predicted melting temperature of amplicons.

under biotic and abiotic stress: *18S rRNA, ACT, BHLH, CLA, CYP, EF1, GAPDH, SAND* and *TIP41*. Their full names, cellular functions and EST GenBank accession numbers are listed in Table 1. From at least three primer pairs per gene designed, the final primer pairs for the nine genes were selected on the basis of their single PCR products, amplification efficiencies and regression coefficients ($R^2$), as shown in Table 2. The primers for each gene amplified a single PCR product at an expected size as confirmed in a 2% agarose gel electrophoresis (Fig. S1) and later sequencing (Seq. S1). Melting curve analysis also confirmed the single PCR products with no primer-dimers (Fig. S2), and their melting temperatures were consistent with the prediction (Table 2). Under optimized reaction conditions (Table 2), their $R^2$ values ranged from 0.992–0.999, and amplification efficiencies ranged from 93.0–105.4%.

## Transcription profiling of the reference genes

The expression levels of the nine candidate reference genes were assessed for expression stability under diverse conditions including different developmental stages and biotic or abiotic stresses. The Cq (quantification cycle) values for the candidate reference genes in all the samples were used to compare the expression levels among these genes (Fig. 1). The data analysis showed that these reference genes were moderately abundant in all tested samples, exhibiting a wide range of expression levels. The mean Cq values of the

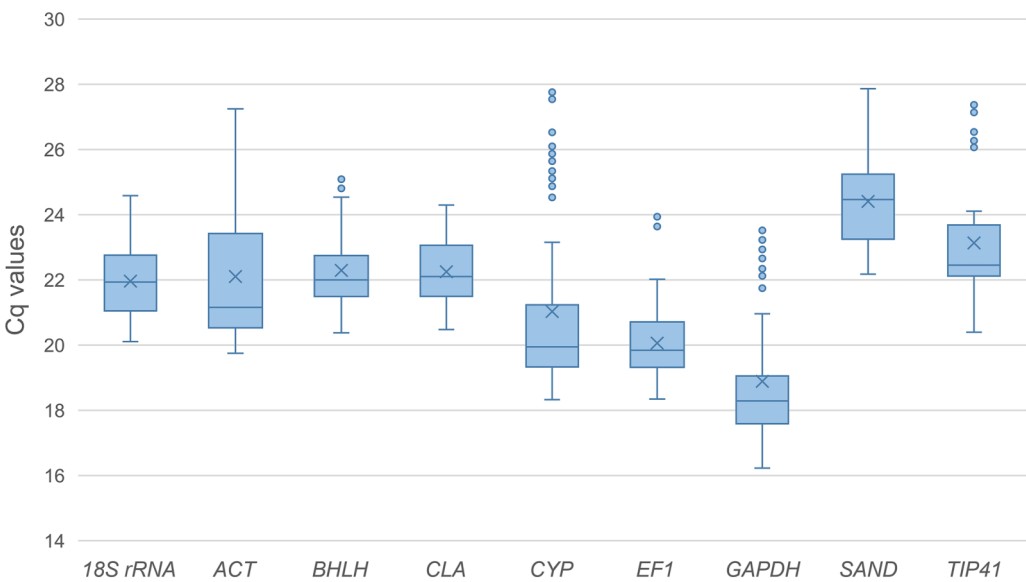

**Figure 1 Quantification cycle (Cq) values of the nine candidate reference genes across all the experimental samples.** Whiskers represent the local maximum and minimum values, and points beyond the end of each whisker mark the outliers. Lower and upper boxes indicate the 25th to 75th percentile, and the midlines inside the boxes represent the medians. The means are denoted by "×" markers.

reference genes varied from 18.88–24.40, with most lying between 20 and 23. *GAPDH* showed the highest expression level in all the samples with the lowest Cq value (18.89), while *SAND* showed the lowest expression level with the highest Cq value (24.41). *CYP, ACT* and *GAPDH* showed great variability with the Cq value ranging from 18.40–27.65, 19.90–26.85 and 16.37–23.29 respectively. Since the expression levels of the nine candidate reference genes showed a variation across all samples, further evaluation would be necessary to identify the best selection(s) from these candidate genes for normalizing gene expression under certain experimental conditions.

## Gene expression stability analysis

Expression stabilities of the nine candidate reference genes were analyzed by geNorm and NormFinder. The geNorm program calculated average expression stability (M) for a reference gene based on the average pairwise variation between all genes tested (*Vandesompele et al., 2002*). The rankings of the nine candidate reference genes under different conditions according to the M values are depicted in Fig. 2. When all the samples were taken together, *BHLH, SAND, CLA* and *EF1* (in order, similarly hereinafter) were the most stably expressed genes, while *CYP* and *18S rRNA* were the least stable ones. In leaves across various developmental stages, *EF1, SAND, TIP41* and *GAPDH* performed well, while *CYP* and *BHLH* were the least stable genes. *CLA, SAND* and *EF1* had the most stable expression under biotic stress, whereas *CYP* and *18S rRNA* were the most variably expressed of the nine candidate genes. Under drought stress, *CYP, SAND, GAPDH* and *TIP41* were the most stable genes. By contrast, the most stable ones for salinity stress were *CLA, TIP41, BHLH* and *GAPDH*. In response to

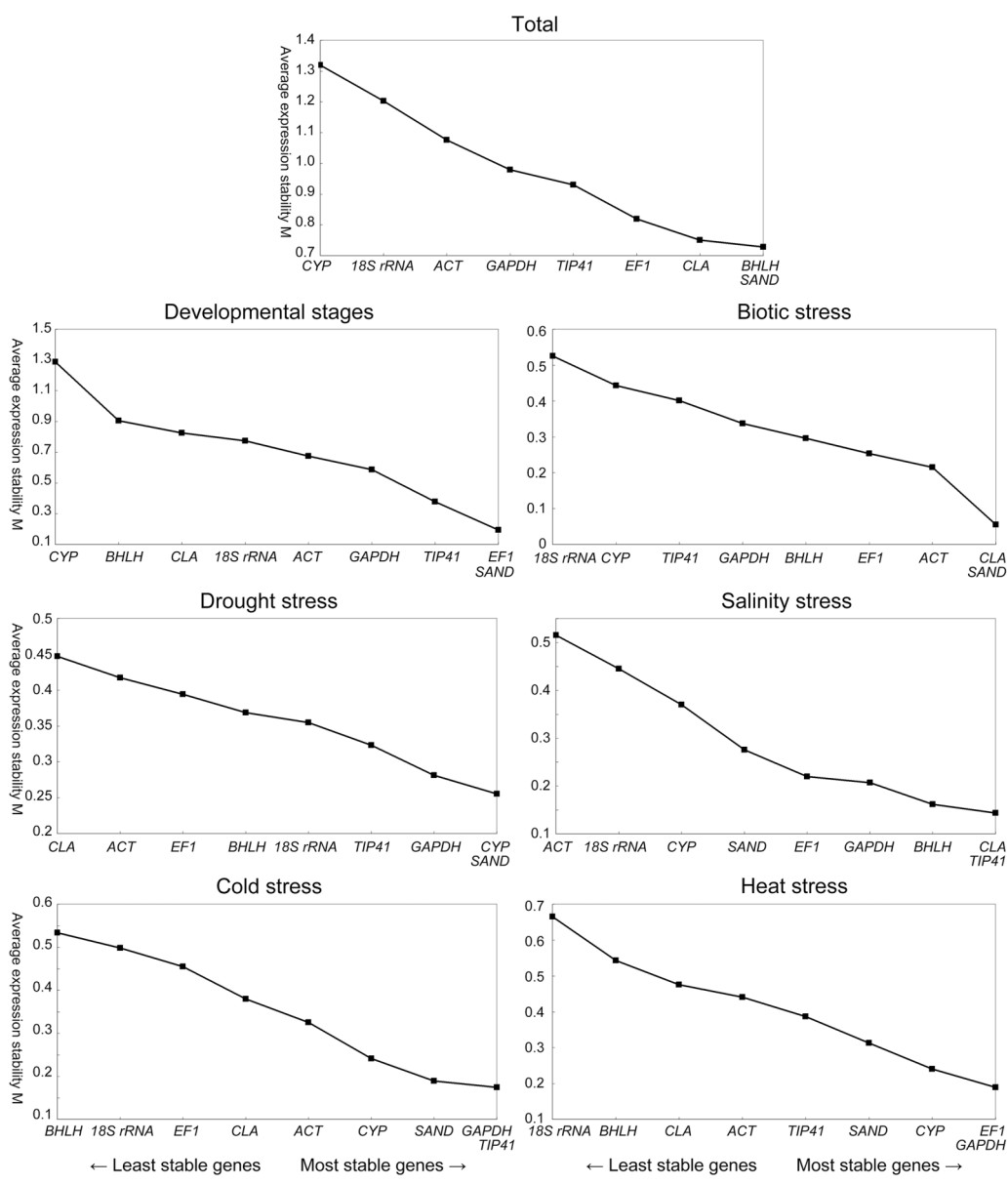

**Figure 2 Average expression stability and ranking of the nine candidate reference genes calculated by geNorm.** A lower value of average expression stability (M) indicates more stable expression. The least stable genes are on the left and the most stable genes on the right.

cold and heat stress, *GAPDH*, *CYP* and *SAND* were identified as the best two reference genes for both treatments.

The geNorm program was also used to determine the optimal number of reference genes required for the normalization across different sets of experiment by calculating pairwise variation ($V_n/V_{n+1}$) value. According to the instruction, 0.15 was suggested to be the cut-off value for determining the optimal number of reference genes, below which the inclusion of additional reference genes is not required (*Vandesompele et al., 2002*). For various developmental stages, the V4/5 value was 0.145 as shown in Fig. 3, which means four reference genes, namely *EF1*, *SAND*, *TIP41* and *GAPDH* should be considered.

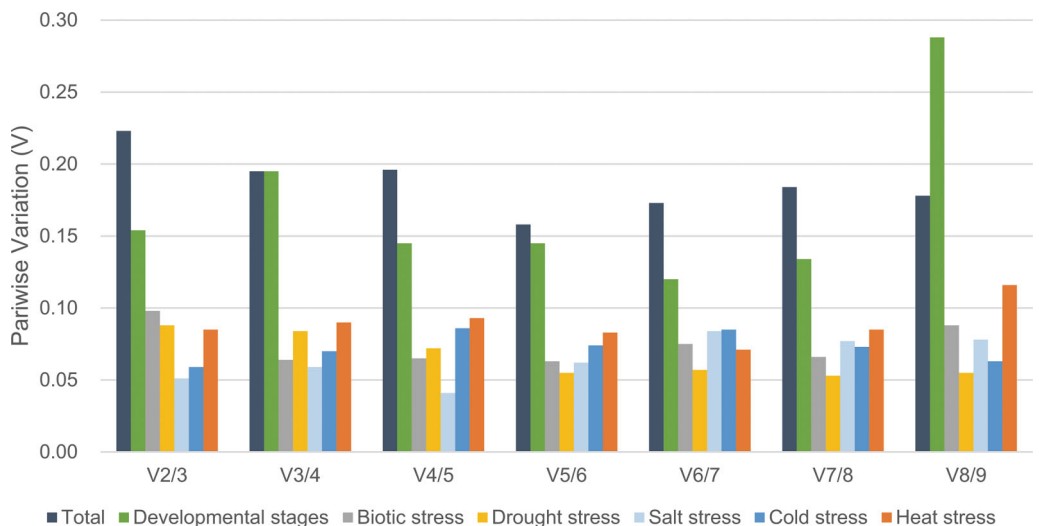

**Figure 3 Determination of the optimal number of reference genes by pairwise variation using geNorm.** Pairwise variation ($V_n/V_{n+1}$) was analyzed between the normalization factors $NF_n$ and $NF_{n+1}$, carried out for all the samples (Total) and the samples at different developmental stages or under different stresses.

The experimental data for biotic stress (V2/3 value = 0.098) showed that *CYP* and *SAND* were sufficient for normalization. For drought stress (V2/3 value = 0.088) and salinity stress (V2/3 value = 0.051), the suitable reference genes were *CYP/SAND* and *CLA/TIP41*, respectively. In addition, the V2/3 value of cold stress was 0.059, indicating that these two reference genes, *GAPDH* and *TIP41*, were sufficient for normalization of gene expression. *EF1* and *GAPDH* produced a V2/3 value of 0.085 in heat stress treatment, suggesting these two candidate genes could be selected as appropriate reference genes. However, the $V_n/V_{n+1}$ values were higher than 0.15 when taken all the samples together, and thus the cut-off value of 0.15 was a little strict in this case (*Fernandez et al., 2011*). The V5/6 value was 0.158, indicating that the optimal number of reference targets would be five: *BHLH, SAND, CLA, EF1* and *TIP41*.

In general, similar results were obtained when the candidate reference genes were analyzed through NormFinder, which is based on variance estimation to calculate the Normalization Factor (NF) (*Andersen, Jensen & Orntoft, 2004*). These genes were ranked according to their stabilities under a given set of experimental treatments (Table 3). The ranking under biotic stress was similar as that of geNorm analysis. *CYP* and *TIP41* were also recognized as the most stable genes under drought and salinity stress, respectively, while differences were observed in the analysis of genes under cold and heat stress, where *SAND* and *CYP* were ranked as the most stable one respectively instead. *TIP41* and *ACT* were the most stable genes for various developmental stages. When all the samples were considered, *TIP41, EF1, BHLH* and *SAND* were the most stably expressed genes suggested by NormFinder, while *CYP* and *18S rRNA* were suggested as the least stable genes by both the algorithms.

Their expression stability of expression was re-analyzed using the BestKeeper algorithm, which provides the Standard Deviation (SD) based on the Cq values of all
**Table 3 Gene expression stability of the nine candidate reference genes under different conditions calculated by NormFinder.**

| Total | | Developmental stages | | Biotic stress | | Drought stress | | Salinity stress | | Cold stress | | Heat stress | |
|---|---|---|---|---|---|---|---|---|---|---|---|---|---|
| Gene | NF | Gene | NF | Gene | NF | Gene | NF | Gene | NF | Gene | NF | Gene | NF |
| TIP41 | 0.382 | TIP41 | 0.158 | CLA | 0.058 | CYP | 0.117 | TIP41 | 0.054 | SAND | 0.052 | CYP | 0.073 |
| EF1 | 0.413 | ACT | 0.268 | SAND | 0.073 | 18S rRNA | 0.139 | EF1 | 0.073 | CYP | 0.090 | TIP41 | 0.164 |
| BHLH | 0.436 | EF1 | 0.269 | ACT | 0.096 | GAPDH | 0.145 | CLA | 0.074 | GAPDH | 0.115 | EF1 | 0.206 |
| SAND | 0.469 | SAND | 0.331 | EF1 | 0.154 | TIP41 | 0.177 | GAPDH | 0.083 | TIP41 | 0.173 | CLA | 0.240 |
| GAPDH | 0.483 | GAPDH | 0.414 | BHLH | 0.215 | BHLH | 0.199 | BHLH | 0.092 | EF1 | 0.329 | GAPDH | 0.282 |
| CLA | 0.657 | 18S rRNA | 0.470 | GAPDH | 0.218 | SAND | 0.258 | SAND | 0.270 | CLA | 0.348 | ACT | 0.315 |
| ACT | 0.768 | CLA | 0.620 | TIP41 | 0.353 | EF1 | 0.265 | CYP | 0.418 | ACT | 0.349 | SAND | 0.362 |
| 18S rRNA | 1.068 | BHLH | 0.915 | CYP | 0.419 | ACT | 0.311 | 18S rRNA | 0.437 | BHLH | 0.383 | BHLH | 0.455 |
| CYP | 1.085 | CYP | 1.789 | 18S rRNA | 0.543 | CLA | 0.333 | ACT | 0.478 | 18S rRNA | 0.385 | 18S rRNA | 0.717 |

**Note:**
A lower normalization factor (NF) value indicates higher stability of the gene.

candidate reference genes, as well as the correlation coefficients (r) of the genes with the BestKeeper Index calculated from the geometric mean of the remaining reference genes (*Pfaffl et al., 2004*). As shown in Table 4, the genes with a SD [± CP] value below 1 and a SD [± x-fold] value below 2 are considered to be qualified for the gene expression normalization, and their r values (correlation coefficient) are used for ranking. According to these criteria, *EF1*, *BHLH* and *CLA* were suggested as the three most stable genes when all the experimental conditions were taken into consideration. During different developmental stages, *EF1*, *SAND* and *CLA* were selected by the program for the most stable normalization. Almost all the reference genes were within acceptable criteria under various stresses, and the results were partially similar with those obtained by NormFinder.

Because of the diverse suggestions made by the three algorithms regarding the most stable reference genes, we used Venn charts to identify the most suitable ones under certain conditions by combining the top three genes for each algorithm together (Fig. 4). All the algorithms agreed that *BHLH* was the most stably expressed gene across all the experimental conditions, while *CLA* and *EF1* were recommended by two of the three programs respectively. As a synthesis of the three programs' output, *EF1* under various developmental stages, *SAND* under biotic stress, *CYP/GAPDH* under drought stress, and *TIP41* under salinity stress were generally considered suitable. All the three algorithms were consistently in agreement on the stability of *SAND* and *GAPDH* under cold stress, while only *CYP* was selected by all the three programs as for heat stress. Notably, *ACT* and *18S rRNA*, the most commonly used reference genes, were hardly found to be among the most stable transcripts as suggested by our results.

## Normalization of *LrLOX*

The expression level of *LrLOX* after inoculation with *B. elliptica* was calculated with the reference genes suggested as above. According to the result of the evaluation for biotic stress, *SAND* was the most stable reference genes, while *CYP* was the least stable one

**Table 4 Descriptive statistics and stability values of the nine candidate reference genes under different conditions calculated by BestKeeper.**

| | | 18S rRNA | ACT | BHLH | CLA | CYP | EF1 | GAPDH | SAND | TIP41 |
|---|---|---|---|---|---|---|---|---|---|---|
| Total | SD [± CP] | **0.794** | 1.826 | **0.982** | **0.799** | 2.048 | **0.866** | 1.508 | 1.045 | 1.374 |
| | SD [± x-fold] | **1.734** | 3.546 | **1.975** | **1.740** | 4.135 | **1.823** | 2.844 | 2.064 | 2.592 |
| | Coeff. of corr. (r) | 0.356 | 0.958 | 0.885 | 0.766 | 0.921 | 0.904 | 0.970 | 0.871 | 0.966 |
| Developmental stages | SD [± CP] | 1.205 | 1.643 | **0.702** | **0.690** | 3.190 | **0.988** | 1.690 | **0.919** | 1.258 |
| | SD [± x-fold] | 2.305 | 3.122 | **1.626** | **1.613** | 9.126 | **1.984** | 3.226 | **1.891** | 2.391 |
| | Coeff. of corr. (r) | 0.909 | 0.993 | 0.693 | 0.899 | 0.940 | 0.967 | 0.967 | 0.952 | 0.996 |
| Biotic Stress | SD [± CP] | **0.257** | **0.334** | **0.235** | **0.353** | **0.604** | **0.261** | **0.376** | **0.366** | **0.594** |
| | SD [± x-fold] | **1.195** | **1.261** | **1.177** | **1.277** | **1.520** | **1.198** | **1.298** | **1.288** | **1.509** |
| | Coeff. of corr. (r) | −0.552 | 0.909 | 0.625 | 0.930 | 0.955 | 0.792 | 0.780 | 0.931 | 0.963 |
| Drought stress | SD [± CP] | **0.140** | **0.429** | **0.267** | **0.405** | **0.361** | **0.196** | **0.413** | **0.492** | **0.405** |
| | SD [± x-fold] | **1.102** | **1.347** | **1.204** | **1.325** | **1.285** | **1.146** | **1.332** | **1.406** | **1.324** |
| | Coeff. of corr. (r) | 0.904 | 0.636 | 0.740 | 0.544 | 0.944 | 0.408 | 0.944 | 0.977 | 0.859 |
| Salinity stress | SD [± CP] | **0.458** | **0.945** | **0.494** | **0.530** | **0.621** | **0.508** | **0.574** | **0.509** | **0.599** |
| | SD [± x-fold] | **1.373** | **1.925** | **1.408** | **1.444** | **1.538** | **1.422** | **1.488** | **1.423** | **1.514** |
| | Coeff. of corr. (r) | 0.603 | 0.974 | 0.964 | 0.966 | 0.762 | 0.975 | 0.975 | 0.850 | 0.977 |
| Cold stress | SD [± CP] | **0.223** | **0.627** | **0.560** | **0.555** | **0.397** | **0.189** | **0.396** | **0.444** | **0.429** |
| | SD [± x-fold] | **1.167** | **1.544** | **1.474** | **1.469** | **1.317** | **1.140** | **1.316** | **1.360** | **1.346** |
| | Coeff. of corr. (r) | 0.005 | 0.821 | 0.778 | 0.875 | 0.936 | 0.328 | 0.930 | 0.972 | 0.898 |
| Heat stress | SD [± CP] | 1.005 | **0.369** | **0.529** | **0.512** | **0.209** | **0.141** | **0.112** | **0.347** | **0.355** |
| | SD [± x-fold] | 2.008 | **1.291** | **1.443** | **1.426** | **1.156** | **1.102** | **1.080** | **1.272** | **1.279** |
| | Coeff. of corr. (r) | 0.846 | 0.511 | 0.382 | 0.972 | 0.937 | 0.378 | −0.412 | 0.223 | 0.786 |

**Note:**
A SD [± CP] value below 1 and a SD [± x-fold] value below 2, marked in bold, suggest that the reference gene is qualified for normalization. The values of correlation coefficient (r) were used for ranking in this study.

among the nine candidate genes. *ACT* and *CLA* were also recommended by two of the three algorithms, but *ACT* was considered to have a relatively lower stability than *CLA* and *SAND*. As shown in Fig. 5, when normalized with *SAND*, the transcription level of *LrLOX* was down-regulated (Fold Change (FC) = 0.65) at 2 hpi, and a maximum level (FC = 2.23) was obtained at 12 hpi (hours post inoculation) relative to uninoculated controls. Upregulation was still observed at 24 hpi, but the mRNA level was then steadily decreased in the next 24 h. Similar expression patterns were generated when normalized with *CLA* separately or in combination with *SAND* together. When normalized using *ACT*, the expression level of *LrLOX* at 24 hpi was shown as down-regulated (FC = 0.87), but it showed no significant difference if compared with the value normalized with *SAND* (Fig. 5). Normalization with a combination of the three most stable genes also made a consistent result. When normalized with *BHLH*, which was considered as the most stable gene overall but was ranked in the middle under biotic stress, the expression pattern of *LrLOX* was still roughly recognizable. However, significant difference could be observed at 12 hpi, as the maximum mRNA level was reduced to 1.70 fold. As expected, the expression profile showed more variation when normalized with *CYP*, the least stable genes calculated by the geNorm. Apart from the incorrect down regulation at 24 hpi here, the upregulation at 12 hpi was even as high as 6.66 fold.

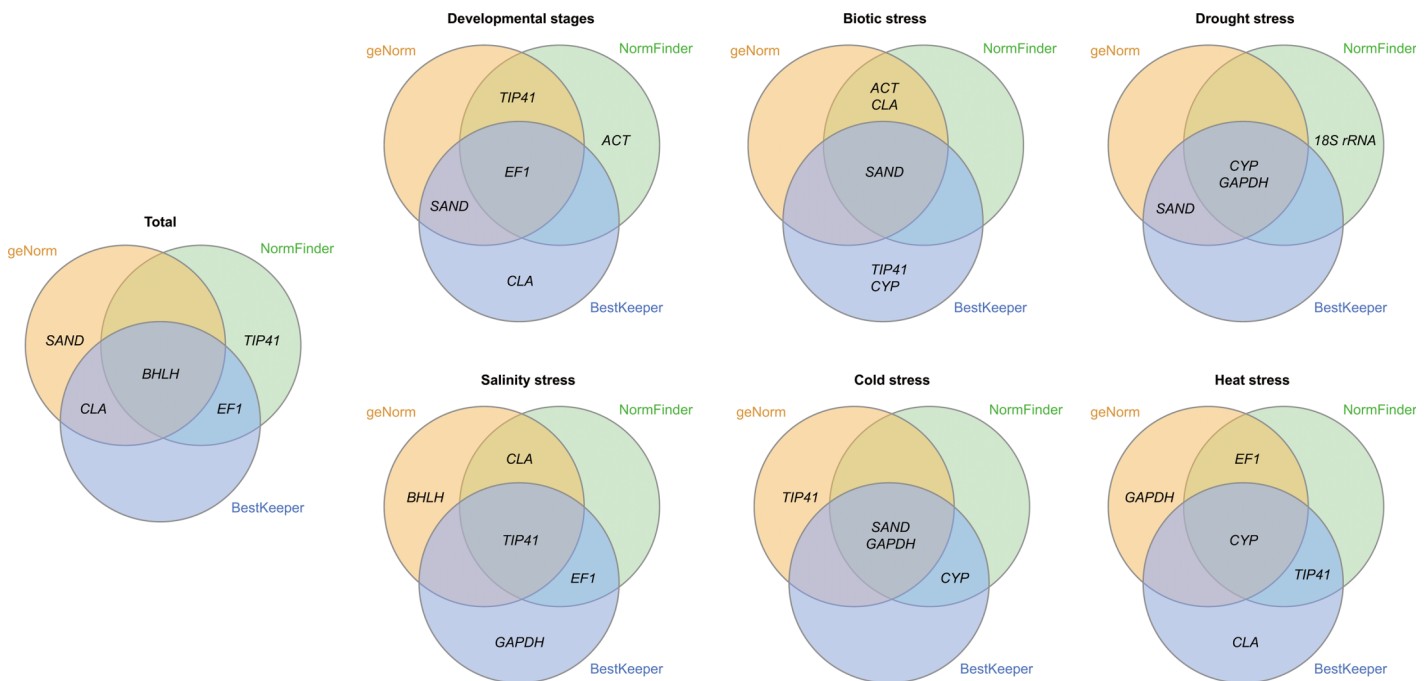

**Figure 4  Venn diagram showing the most stable genes identified by geNorm, NormFinder and BestKeeper.** The three most stable genes selected by each algorithm are presented.

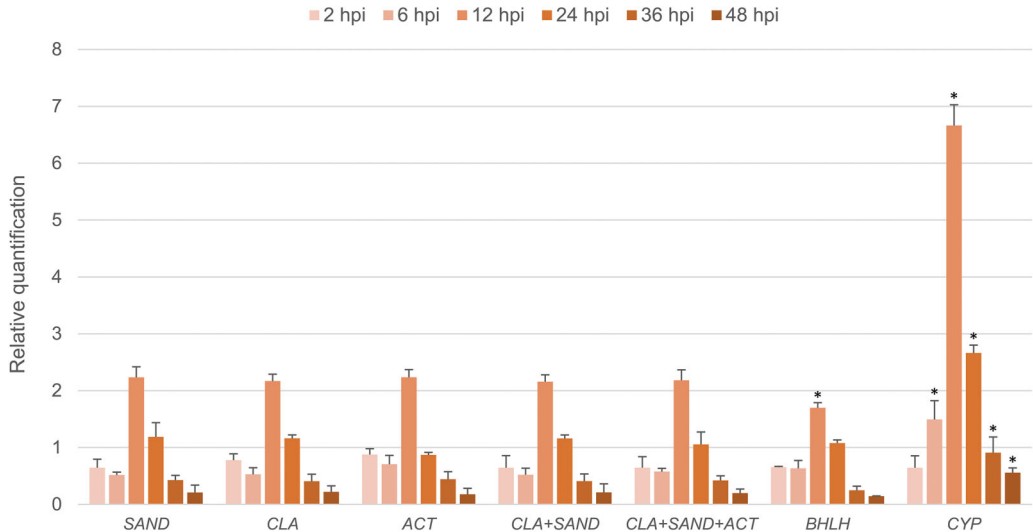

**Figure 5  Relative quantification of *LrLOX* in leaves at different time points after inoculation with *B. elliptica*.** Asterisks (*) indicate significant differences ($p < 0.05$) relative to the value normalized with *SAND* at each time point.

## DISCUSSION

The accuracy of normalization in qPCR is directly relied on the expression stability of the reference genes (*Gachon, Mingam & Charrier, 2004*). Thus, the selection of stable reference gene for a wide range of conditions is crucial for quantification of gene

expression. In our study, three popular statistical algorithms, BestKeeper, geNorm and NormFinder, were used for evaluation of stable reference genes in *L. regale.* NormFinder estimates both the intra- and inter-group variation and combines them into a stability value (*Andersen, Jensen & Orntoft, 2004*), and GeNorm evaluates the stability of reference gene by pairwise comparison of expression ratio variation of among reference genes in specific experimental sets (*Vandesompele et al., 2002*). These two algorithms both correct for inter-sample variation, but BestKeeper do not account for differences in RNA quality/input or Reverse Transcription (RT) efficiency across samples (*Jiang et al., 2014*). According to the results, the suitable genes under each condition selected by these algorithms share much in common. Moreover, geNorm and NormFinder were also in agreement on the least stable gene for each treatment. The accordance in the results provided by these algorithms was also mentioned in some previous study on other plants (*De Spiegelaere et al., 2015*; *Mafra et al., 2012*). However, obvious divergence in these results still exists in some situations, mainly caused by the varying priorities in different algorithms. BestKeeper even provides multiple factors in its result, thus different sort criteria would also affect the final ranking. Since every algorithm has its own risks and benefits for the evaluation, it is generally accepted to combine these results together for optimal selection of suitable reference genes (Fig. 4).

Several studies involving the gene expression in *Lilium* have been reported recently. However, due to the absence of an extensive evaluation of the reference genes in *Lilium*, the selection of them was only based on previous experience in other plants rather than a practical validation on their suitability in the genus. The selection of reference gene(s) might influence the results or even conclusion of studies, especially those concerning plant development or stress response, as there would be more variation in gene expression under those conditions. As a commonly used reference gene, the *ACT* gene family associated with cell structure maintenance was previously used as the reference gene for normalization in *Lilium* (*Wang et al., 2009*; *Xin et al., 2010*), and it showed stable expression in *L. davidii* var. *unicolor* (*Li et al., 2015*). Nevertheless, in our study, *ACT* failed to be ranked as a satisfactory reference gene under most of the experimental conditions except biotic stress, and even yielded the poorest values during drought and salinity stress. On the other hand, *18S rRNA* was previously proved to be one of the best reference genes for *L. brownii* and some other plant species (*Kianianmomeni & Hallmann, 2013*; *Luo et al., 2014*; *Saha & Vandemark, 2013*), but in our study, *18S rRNA* was generally evaluated as the one of the least stable reference genes except under drought stress. This is possibly caused by their genotype differences. Some studies on other species also recognized it as an unstable gene (*Castro et al., 2012*; *Qi et al., 2010*). *GAPDH* was used for normalization in some gene expression studies concerning the resistance of *L. regale* to *Fusarium oxysporum* (*He et al., 2014*; *Li et al., 2014*), but our study revealed that it was not stable enough after the plant was inoculated with another important pathogenic fungus, *B. elliptica*, despite its good stability under abiotic stress. Therefore, it would be helpful for future expression studies on *Lilium*, particularly *L. regale*, to take other reference genes into consideration apart from these regular choices, as our study suggested.

In general, *BHLH* was recognized as one of the most stably expressed reference genes in all the samples and experimental treatments, but interestingly, it failed to have a good performance under specific conditions. A similar result concerning reference gene *psaA* was also found in the study on chrysanthemum (*Gu et al., 2011*). As a gene encoded a helicase, basic helix-loop-helix family protein, *BHLH* was reported to have a stable expression in model plants such as *Arabidopsis thaliana* (*Czechowski et al., 2005*) and *Medicago truncatula* (*Kakar et al., 2008*). Because there is actually not an absolutely stable reference gene in all cases, *BHLH* might be a compromise choice rather than a satisfying one. The normalization of *LrLOX* also confirmed that the result using *BHLH* was not so accurate as those using the recommended genes. It also should be noted that *SAND* acted as a stably but quite lowly expressed gene when compared with other reference genes (Fig. 1). *SAND* might be therefore recommended for the quantification of low abundance transcripts such as transcription factors, so that the difference in quantification cycle ($\Delta$Cq) values between the reference transcript and the gene of interest will then be much smaller, and the results will be less influenced by variations in amplification efficiencies and hence more accurate.

As mentioned above, the most suitable genes still varied significantly depending on experimental conditions. In the present study, *EF1* was suggested as the most stable reference genes during development in lily leaves, whereas it is reported that *EF1* is unsuitable for normalization at different developmental stages in oil palm (*Elaeis guineensis*) (*Yeap et al., 2014*). For biotic stress, *SAND* was indicated as the most stable reference genes by all the three algorithms. A combination of two reference genes, *CLA/SAND*, comprised the optimum set under biotic stress. Studies on *Humulus lupulus* and *Citrus* also revealed a stable expression level of *SAND* (*Mafra et al., 2012*; *Stajner, Cregeen & Javornik, 2013*), but our result is different from the work on lentil (*Lens culinaris*) and soybean (*Glycine max*), in which *18S rRNA* and *Actin* were the best ones (*Ma et al., 2013*; *Saha & Vandemark, 2013*). According to our results, *CYP* and *TIP41* were considered to express constantly under drought and salinity stress respectively, and *GADPH* was also relatively stable under both the stresses. Under cold stress condition, a previous study on lentil showed that *GAPDH* was the least stable gene (*Saha & Vandemark, 2013*), but in our study on *Lilium*, *GAPDH* was calculated as the one of the most stable genes. On the other hand, *GAPDH* was expressed at a more constant level during heat stress treatment in *Chrysanthemum* (*Gu et al., 2011*), but *CYP* would be a better choice for *Lilium* according the combination of three algorithms. Overall, our results emphasize the importance to identify the most suitable reference genes for individual organisms and different experimental conditions.

Lipoxygenases (LOXs), a family of non-heme-iron-containing fatty acid dioxygenases, play a crucial role in lipid peroxidation processes during plant defense responses to biotic stresses such as pathogen infection (*Hwang & Hwang, 2010*). LOXs catalyze the oxidation of polyunsaturated fatty acids into oxylipins, which are then enzymatically metabolized into traumatin, Jasmonic Acid (JA) and Methyl Jasmonate (MeJA) (*Joo & Oh, 2012*). These compounds are considered to respond to plant development, senescence and diverse stresses. To evaluate the reference genes selected in this study,

we analyzed the expression level of *LrLOX* in the leaves of *L. regale* infected with *B. elliptica*. The expression model of *LrLOX* in this study was concordant with that of *PvLOX6* in *Phaseolus vulgaris* after infected with *Pseudomonas syringae* (*Porta, Figueroa-Balderas & Rocha-Sosa, 2008*). The expression profile of *LrLOX* was hardly considered to be reliable when referring to the unstable genes concluded under biotic stress. Our results showed that reference genes would have a severe impact on the expression level of the gene of interest, and the selection of them should be based on a careful and comprehensive evaluation. Therefore, our study should be a new start for the gene expression-based studies on *L. regale*, and would be beneficial for the molecular breeding of *Lilium* in the future.

## CONCLUSION

In summary, we evaluated the expression stability of nine candidate reference genes in *L. regale* under different developmental stages, biotic and abiotic stresses to identify the stable one(s) for normalization in gene expression studies. Based on the combination of the analyses by three algorithms, *BHLH* was considered as the most generally stably expressed reference genes under all the experimental conditions, but specific selection still depended on certain experimental environment. The selection of reference genes under biotic stress was further validated in the investigation on the expression pattern of *LrLOX*. With the results outlined in this study, it would be more feasible to implement a sensitive and accurate quantification of gene expression in *L. regale*.

## ACKNOWLEDGEMENTS

We would like to thank Miss Yu-Qian Zhao and Miss Xue Gao for their helping in our study.

### Funding

This work was supported by the Special Fund for Forest Scientific Research in the Public Welfare (No. 201204609), and the National High Technology Research and Development Program of China (863 Program) (No. 2013AA102706). The funders had no role in study design, data collection and analysis, decision to publish, or preparation of the manuscript.

### Grant Disclosures

The following grant information was disclosed by the authors:
Special Fund for Forest Scientific Research in the Public Welfare: 201204609.
National High Technology Research and Development Program of China (863 Program): 2013AA102706.

### Competing Interests

The authors declare that they have no competing interests.

## Author Contributions

- Qiang Liu conceived and designed the experiments, performed the experiments, analyzed the data, wrote the paper.
- Chi Wei performed the experiments, analyzed the data, wrote the paper, prepared figures and/or tables.
- Ming-Fang Zhang contributed reagents/materials/analysis tools.
- Gui-Xia Jia conceived and designed the experiments, contributed reagents/materials/analysis tools, reviewed drafts of the paper.

## DNA Deposition

The following information was supplied regarding the deposition of DNA sequences:

GenBank: HQ686070, KJ543460, KJ543467, KJ543465, KJ543464, KJ543461, KJ543468, KJ543463, KJ543466, KM051414.

## Data Deposition

The raw data in this study were supplied as Supplemental Dataset file.

## Supplemental Information

Supplemental information for this article can be found online at http://dx.doi.org/10.7717/peerj.1837#supplemental-information.

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
