# Peer review of "Evaluation of putative reference genes for quantitative real-time PCR normalization in Lilium regale during development and under stress"

_PeerJ, doi:10.7717/peerj.1837_

## Round 0.1 · original submission · Minor Revisions

· Academic Editor

Minor Revisions

With your manuscript you present original primary research within the scope of PeerJ. You robustly validated a set of qPCR reference genes for an important floriculture crop. Your conclusions are valid while the manuscript is very well written and easy to follow. However, the two reviewers have found some points that require some changes and minor revisions before we could accept your manuscript form publication.

Reviewer 1 ·

Basic reporting

Please provide RPKM values for the nine selected genes as the addition to table 1 to understand selection process and compare with qPCR results. Please provide also predicted Tm of the amplicons for each gene as the addition to t able 2 so it could be verified with the experimental data presented on FigS2. Please provide sequencing results in supplementary figure.

Experimental design

No Comments

Validity of the findings

Please provide statistical significance for the results presented on figure 5. Differences in FC discussed in the paper are relatively small and error bars seems to be overlapping. Expression patterns for LrLOX are fairly similar even when using ACT as the reference gene. I would suggest using one-way ANOVA and Tukey's range test. The only clear conclusion could be drawn about inaccuracy of the normalisation vs CYP reference gene.
SAND gene should be highlighted as a stably but quite lowly expressed gene (4 fold lower than Actin/18S rRNA/BHLH and 64-fold lower than GAPDH .) SAND might be therefore reference of choice for quantification of low abundance transcripts such as transcription factors. Because the difference in threshold cycle number (Delta Cq) between the reference transcript and the gene under investigation (e.g. a transcription factor transcript) will then be much smaller, the calculation of the results will be less influenced by variations in amplification efficiencies and hence more accurate.

Additional comments

No Comments

Reviewer 2 ·

Basic reporting

I havent got any special comments related with "basic Reporting"

Experimental design

There are a few limitations that should be added to the manuscript " see general comments to the authors"

Validity of the findings

The data are robust and the conclusions adequately supported by the data

Additional comments

Recently, the validation and selection of suitable reference genes for accurate normalization of gene expression data in plants is receiving much more attention. This paper attempts to identify appropriate reference genes for gene expression studies in Lillium regale. The stability of nine potential reference genes was evaluated throughout development and a wide range of experimental conditions, including biotic and abiotic stresses. The expression stability of these genes was analyzed using geNorm, NormFinder and BestKeeper softwares in order to identify the best reference genes for use under given experimental conditions. Additionally, the most stably expressed reference genes were used for accurate normalization of the expression level of LrLOX in leaves of Lillium regale inoculated with B. elliptica.
It seems that one of the conclusions of the study was that none of the candidate reference genes was uniformly expressed across development and all the experimental conditions tested in this study, although BHLH was considered the most stably expressed. In addition, some of the most frequently used reference genes as ACT and 18S rRNA do not always have stable transcript levels and their use may compromise gene expression. The work advances in the validation of reference genes in an important ornamental plant and can make a worthy contribution for the scientific community working with this plant species.

The paper may be suitable for publication after some revision.
Although I consider this is a well conducted study. There are a few limitations that need to further discussion. The first is that the information related with growth conditions is limited: It just mentioned the length of photoperiod (12 h light/12 h dark) and temperature (25 °C day/22 °C night), what about other climate parameters as the relative humidity and light intensity used in this study?.
It is mentioned in line 93 “For drought stress treatment, the water supply was withheld and leaf samples were collected at 2, 3, 4, 6 and 8 d during the treatment some additional”. That means the rest of the plants were daily watered. Such information should be added to this section (in the first paragraph of plant material section).
Was the watering to field capacity?.
How was carried out the incubation of the inoculated leaves in the growth chamber?

I will also suggest a proofreading of the manuscript as long as certain information was obscured by writing difficulties as:

Abstract
Lines 16-17 “which has been widely used for quantitative measurements of gene expression”. Does it mean? for quantification of gene expression or for transcript quantification.

Introduction
Line 44 “or the treatments introduced by experiments” it could be “or the experimental conditions”

Line 57 “the optimal gene among a set of candidates for qPCR normalization”. May be is more accurate to use the most suitable gene or the best gene among a set of candidates…

Lines 64-65. The statement “the reference genes used in the studies..” should be replace by “the reference genes used in these studies”

Lines 79-80 Replace “in this study, the stabilities of nine putative reference genes were evaluated” by in this study, the expression stability of nine putative reference genes were evaluated
Materials and Methods

Lines 126-127 Replace “Primers were designed using Beacon designer 7 (Premier Biosoft, USA) using the following criteria” by “Primers were designed using Beacon designer 7 (Premier Biosoft, USA) following the stringent criteria”

Line 212 Replace “two reference genes for both the treatments” by “two reference genes for both treatments”

Line 225 The sentence “these two candidate genes could be the choice of optimal reference genes” means “these two candidate genes could be selected as appropriate reference genes”

Line 233 The sentence “The ranking under biotic stress was same as the result of geNorm analysis” means “The ranking under biotic stress was similar as that of geNorm analysis”

Line 234 Replace “CYP and TIP41 were also recognized as the most stable gene” by “CYP and TIP41 were also recognized as the most stable genes”

Line 235 Introduce “,” after salt stress

Line 241 Replace “Their stabilities of expression was then re-analyzed using the BestKeeper algorithm” by “Their expression stability was re-analyzed using the BestKeeper algorithm”

Line 253 Replace “under certain of conditions” by “under certain conditions”

Discussion

Line 286-287 Replace “Thus the selection of stable reference gene for certain of conditions plays an indispensable role in quantification of gene expression” By “Thus, the selection of stable reference genes for a wide range of conditions is crucial for quantification of gene expression”

Lines 307-310 The phrase “The uncertainty of the applied reference genes might exert an effect on the results or even conclusion of the studies on Lilium, especially studies concerning plant development or stress response, which might bring about expression variation in these reference genes” is awkward

Lines 318-320 The phrase “This is possibly caused by their genotype differences, and the result is also in coordinate with some studies on other species” is awkward

Lines 332-336 The sentence “In view of its relatively low variation in Cq values across different samples (Fig. 1), BHLH might be a compromised choice rather than a satisfying one, as there is actually not an absolutely stable reference gene in all cases. The normalization of LrLOX also confirmed that the result using BHLH was still roughly acceptable, although not so accurate as those using the recommended genes” is awkward

---

## Round 0.2 · accepted · Accept

· Academic Editor

Accept

All points raised by previous reviewer were fully addressed.

Reviewer 1 ·

Basic reporting

No Comments

Experimental design

No Comments

Validity of the findings

No Comments

Additional comments

All of the points raised in my pervious review were properly and fully addressed by the authors. Quality of the manuscript improved to the standard which is ready for publication.